# Signaling cascades and the importance of moonlight in coral broadcast mass spawning

Paulina Kaniewska[1,2†], Shahar Alon[3†‡], Sarit Karako-Lampert[4], Ove Hoegh-Guldberg[5*], Oren Levy[4*]

[1]Australian Institute of Marine Science, Queensland, Australia; [2]School of Biological Sciences, The University of Queensland, St Lucia, Australia; [3]George S Wise Faculty of Life Sciences, Department of Neurobiology, Tel Aviv University, Tel Aviv, Israel; [4]Mina and Everard Goodman Faculty of Life Sciences, Bar-Ilan University, Ramat Gan, Israel; [5]Global Change Institute and ARC Centre of Excellence for Coral Reef Studies, The University of Queensland, St Lucia, Australia

**Abstract** Many reef-building corals participate in a mass-spawning event that occurs yearly on the Great Barrier Reef. This coral reproductive event is one of earth's most prominent examples of synchronised behavior, and coral reproductive success is vital to the persistence of coral reef ecosystems. Although several environmental cues have been implicated in the timing of mass spawning, the specific sensory cues that function together with endogenous clock mechanisms to ensure accurate timing of gamete release are largely unknown. Here, we show that moonlight is an important external stimulus for mass spawning synchrony and describe the potential mechanisms underlying the ability of corals to detect environmental triggers for the signaling cascades that ultimately result in gamete release. Our study increases the understanding of reproductive chronobiology in corals and strongly supports the hypothesis that coral gamete release is achieved by a complex array of potential neurohormones and light-sensing molecules.

**\*For correspondence:** oveh@uq.
edu.au (OHG); oren.levy@biu.ac.il
(OL)

[†]These authors contributed
equally to this work

**Present address:** [‡]Media Lab,
Massachusetts Institute of
Technology, Cambridge, United
States

**Competing interests:** The
authors declare that no
competing interests exist.

**Reviewing editor:** Sonja Pyott,
University Medical Center
Groningen, Netherlands

## Introduction

Adaptation to environmental cycles including daily, tidal, lunar, or annual cycles over millions of years, has enabled marine organisms to synchronize many aspects of their biology, thereby creating important biological rhythms for coping with a variable world (*Tessmar-Raible et al., 2011*). One of nature's greatest examples of synchronized behavior is the coral spawning that occurs on the Great Barrier Reef (GBR). The 'mass-spawning events' on the GBR is the Earth's largest reproduction event. During these annual events (*Babcock et al., 1986*; *Harrison et al., 2011)* changes in the intensity of moonlight trigger the spawning of more than 130 species of scleractinian corals as well as hundreds of other invertebrates over a couple of nights (*Babcock et al., 1986*; *Harrison et al., 2011*). Broadcast spawning requires coral colonies to carefully synchronize the release of egg and sperm into the water column in order to optimize fertilization success (*Harrison et al., 2011*; *Levitan, 2005*). Environmental factors, such as sea surface temperature, lunar phase and the daily light cycle, influence reproductive timing in coral and induce spawning (*Babcock et al., 1986*; *Babcock et al., 1994*; *Harrison et al., 1984*). However, the mechanism by which corals fine-tune and coordinate spawning remains unclear.

Seasonal patterns are crucial for inducing gametogenesis and spawning in broadcast spawning species (*Mendes and Woodley, 2002*), where rising sea temperature is the most likely environmental driver behind the maturation of eggs and sperm (*Babcock et al., 1986*; *Harrison et al., 1984*;

**eLife digest** Sexual reproduction in corals is possibly the most important process for replenishing degraded coral reefs. Most corals are "broadcast spawners" that reproduce by releasing their egg cells and sperm cells into the sea water surface. To maximize their chances of reproductive success, most coral in the Great Barrier Reef – over 130 species – spawn on the same night, during a time window that is approximately 30-60 minutes long. This is the largest-scale mass spawning event of coral in the world, and is triggered by changes in sea water temperature, tides, sunrise and sunset and by the intensity of the moonlight.

How corals tune their spawning behavior with the phases of the moonlight was an unanswered question for decades. Now, Kaniewska, Alon et al. have exposed the coral *Acropora millepora* – which makes up part of the Great Barrier Reef – to different light treatments and sampled the corals before, during and after their spawning periods. This revealed that light causes changes to gene expression and signaling processes inside cells. These changes are specifically related to the release of egg and sperm cells, and occur only on the night of spawning.

Furthermore, by exposing corals to light conditions that mimic artificial urban "light pollution", Kaniewska, Alon et al. caused a mismatch in certain cellular signaling processes that prevented the corals from spawning. Reducing the exposure of corals to artificial lighting could therefore help to protect and regenerate coral reefs.

Future work will involve comparing these results with information about a coral species from another part of the world to investigate whether there is a universal mechanism used by corals to control when they spawn.

*Oliver et al., 1988*). The phase of the moon, on the other hand, coordinates the timing of mass spawning, selecting neap tides that reduce gamete dilution in the water column (*Babcock et al., 1986*; *Babcock et al., 1994*; *Mendes and Woodley, 2002*; *Willis et al., 1985*). Spawning occurs a few days after the full moon and at a precise time after sunset, although the time of day and night(s) on which spawning and gamete release occurs is highly species-specific. In addition to being influenced by moonlight (*Babcock et al., 1986*; *Harrison et al., 1990*; *Glynn et al., 1991*), the timing of spawning is thought to be regulated by the onset of darkness (*Babcock et al., 1986*), the duration of regionally calm weather (*van Woesik, 2010*), food availability (*Fadlallah, 1981*), twilight chromaticity (*Sweeney et al., 2011*), and salinity (*Jokiel, 1985*). How these exogenous factors function together with endogenous mechanisms in corals to achieve mass spawning harmonization is unknown.

## Results and discussion

To explore the interaction of exogenous and endogenous factors, we characterized the transcriptome from 16 *Acropora millepora* colonies over 8 days leading up to the spawning night, as well as during and after spawning (*Figure 1*). These results were compared with transcriptomes of four *A. millepora* colonies sampled 3 months prior to the month of spawning, during full and new moon days at midday and night (12:00, 18:00 and 24:00). A comparison of new and full moon samples, when spawning did not occur, revealed changes in gene expression according to the day within the month, and relative to levels of moonlight (*Figure 1—figure supplement 1*, *Supplementary file 1,2*), thereby demonstrating a response by RNA transcription to the lunar cycle. Notably, many of the genes showing higher expression levels at midnight on full moon days compared with midnight on new moon days, were linked to rhythmic processes (circadian clock related genes) and light signalling, including cryptochrome 1, cryptochrome 2, and thyrotroph embryonic factor.

To elucidate the role of natural moonlight phases and the effect of 'light pollution' on mass spawning synchronization, 16 large *A. millepora* colonies were collected from the Heron Island reef flat (GBR, Australia) and transferred to large outdoor aquaria, which were exposed to natural sunlight, moonlight and flow-through seawater from the reef flat. Coral colonies were exposed to one of the following treatments: ambient (maintained under ambient conditions of daylight and moonlight, N=6), light (illuminated using artificial light, low intensity $\sim 5$ µmole quanta m$^{-2}$ s$^{-1}$ daily from

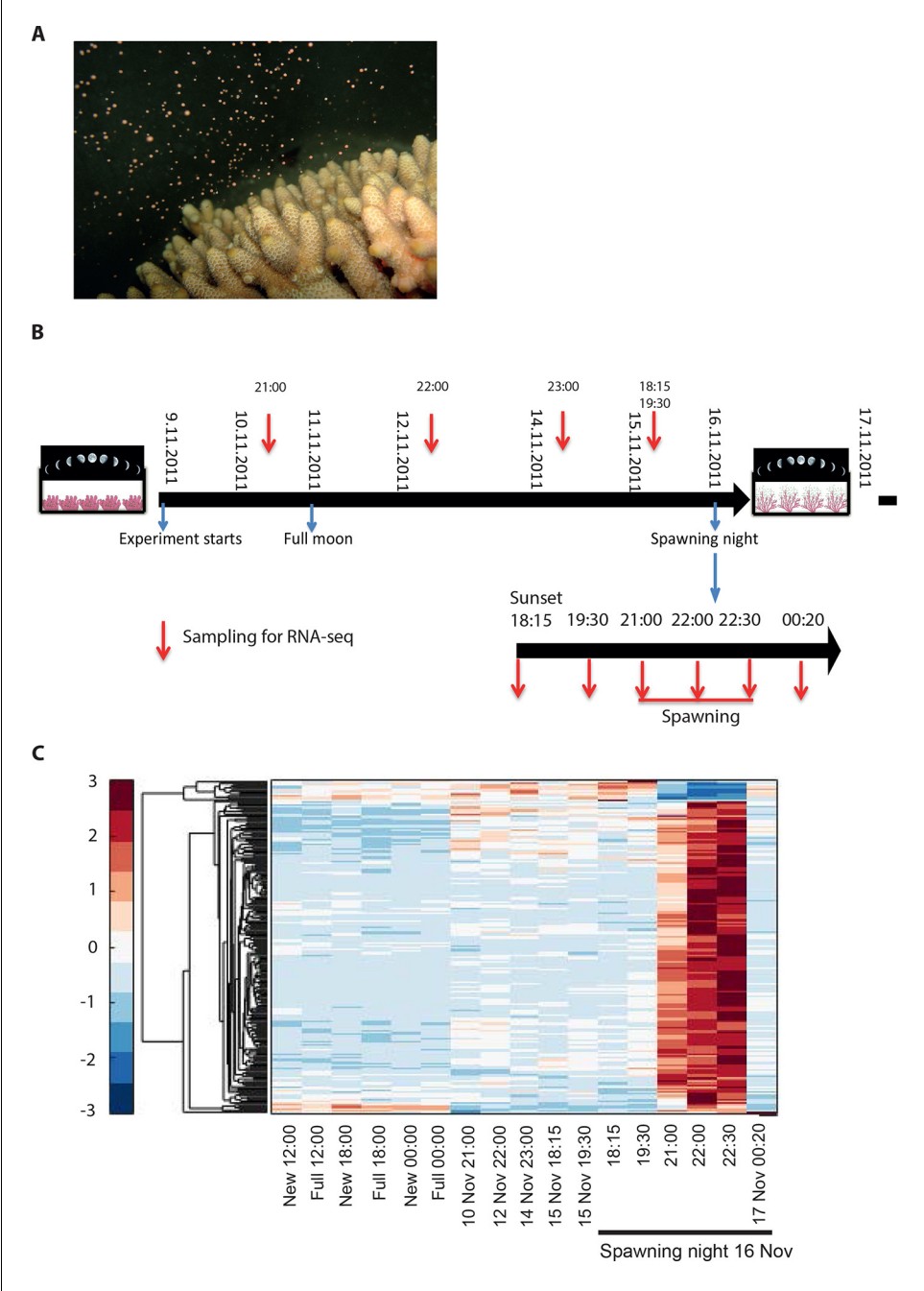

**Figure 1.** Changes in gene expression during mass spawning of the coral, *Acropora millepora*. (**A**) *A. millepora* colony releasing gametes. (**B**) Schematic representing the sampling regime of coral branches from *A. millepora* colonies (colonies were sampled for 8 days leading up to the spawning night, during the spawning night and 1 day following the spawning event), blue arrows indicate timing of experiment start, full moon and observed spawning. Additionally, colonies were sampled prior to the November spawning month (the colonies were sampled at 12:00, 18:00 and 24:00 in August on new moon and full moon days). (**C**) Hierarchical clustering of *A. millepora* gene expression data for the 184 coral transcripts that were only variable during the spawning day using sampling points in August and November. 'New' and 'Full' denote new moon and full moon conditions, respectively.

The following figure supplements are available for Figure 1:

**Figure supplement 1.** Hierarchical clustering of *Acropora millepora* gene expression during full moon versus new moon days at 12:00, 18:00 and 00:00.

**Figure supplement 2.** Hierarchical clustering of the expression patterns of genes that are both expressed and non-constant (Materials and methods), indicates that a sizable number of genes are highly variable only on the spawning day.

sunset to midnight, then kept in the dark until sunrise, N=5), or dark (covered with a black shade from sunset to sunrise to block moonlight, N=5) (*Figure 2A*) beginning 8 days prior to the spawning night. We sampled corals from all treatments during the days leading up to the spawning event at midday, at moonrise and at hourly intervals during the spawning night after sunset; this included sampling of released gametes (*Figure 1B*). During the spawning night, colonies exposed to ambient conditions spawned in a similar manner to those on the reef (gametes were released at approximately 21:30–22:30); however, no sign of spawning behavior occurred in either the light or dark treatments. For ambient corals, we identified 184 transcripts that were only variable during the spawning day and showed a clear change in expression (mostly induction) just prior to and at the time of gamete release (*Figure 1C*); however, such changes were not observed in corals that were kept in the dark (*Figure 2B*). Colonies exposed to the light treatment showed a mismatch of gene expression profiles, and the expression of the genes that were highly variable during spawning occurred prematurely (around sunset at 18:15 and 19:30, *Figure 2B*). Moreover, these colonies did not release gametes, implying that 'light pollution' disrupted spawning synchronization. Gene enrichment analysis revealed that coral transcripts, which were only variable during spawning, were enriched in Gene Ontology (GO) processes of the cell cycle, GTPase activity and signal transduction (*Figure 3A*). An analysis of transcripts that were highly up-regulated or down-regulated during spawning showed a larger proportion of transcripts (177) that increased their expression levels during spawning (*Figure 2—figure supplement 1A*), a substantially smaller portion of 29 transcripts that were down-regulated (*Figure 2—figure supplement 1B*) and no correlation with gene expression profiles from the released gametes. Fifty four out of the 177 up-regulated genes, and none of the 29 down-regulated genes, were also found to be variable during the spawning day (*Supplementary file 1*). The up-regulated coral genes were enriched in G protein-coupled processes, signal transduction and respiratory processes (*Figure 3B*). Down-regulated genes were enriched in processes which generate and maintain rhythms in the physiology of organisms (rhythmic processes) such as circadian clock related genes (*Figure 3C*) and included some additional genes, which were not clustered. A qPCR validation (*Supplementary file 3*) of both up-regulated and down-regulated genes showed strong correlation between the qPCR and RNA-seq data based on $\log_2$ fold change measured on the spawning night in ambient conditions at 19:30 versus 22:00 ($r^2$=0.92, p<0.0001) (*Figure 2—figure supplement 2*).

Since corals do not have specialized visual structures, light detection is likely mediated through photosensitive molecules such as opsins or cryptochromes (*Reitzel et al., 2013*) that help the corals to adapt and to be synchronized with the external irradiance levels (*Panda, 2002*). Furthermore, specific classes of G-proteins can be activated by opsins in response to light in coral larvae (*Mason et al., 2012*). Neuropeptides are another diverse class of signalling molecules that act through G-protein coupled receptors. These peptides have been implicated in an array of biological processes such as reproduction, metabolism, feeding, circadian rhythms, adaptive behaviors and cognition (*Grimmelikhuijzen and Hauser, 2012*). The importance of neuropeptides in life phase transitions has been recorded across a range of taxa such as annelids, cnidarians, insects and mammals (*Schoofs and Beets, 2013*). Our results indicate that gamete release from a broadcast spawning coral is potentially controlled by a series of G-protein-coupled signalling pathways and that the trigger for release might be a non-visual ocular photoreceptor such as the melanopsin-like coral homolog and/or a range of neuropeptide candidates (*Figure 4*, *Supplementary file 1*). Genes induced during spawning included two melanopsin (opn4b)-like homologs, a battery of G-protein-coupled receptors (GPCRs), and a homolog of synaptotagmin 7 (*Figure 3*, *Figure 4*, *Supplementary file 1*). Melanopsin (Opn4) has been shown to be required for light-induced circadian phase shifting in mammals (*Panda, 2002*). Given the broad changes in expression patterns in GPCRs and their potential triggers, it seems that neuropeptide signalling appears to be central in controlling gamete release in broadcast coral spawning, where such a simple nervous system likely evolved in an ancestor common to all cnidarians (*Grimmelikhuijzen and Hauser, 2012*). Our data are consistent with the activation of GPCR receptors and consequent GPCR signalling cascades followed by changes in cell migration (eg, RAP2B, UNC5D, IF2B1), immunity/cell death (eg. TLR2, TLR6, TRAF5, DAB2P, FAK1, NLRC5) respiration (NU4M, NU5M, COX1, NDUA4, CYB) and cytoskeletal organization (eg. PAXI, INF2, NUP43) which together lead to synchronized gamete release (*Figure 4*).

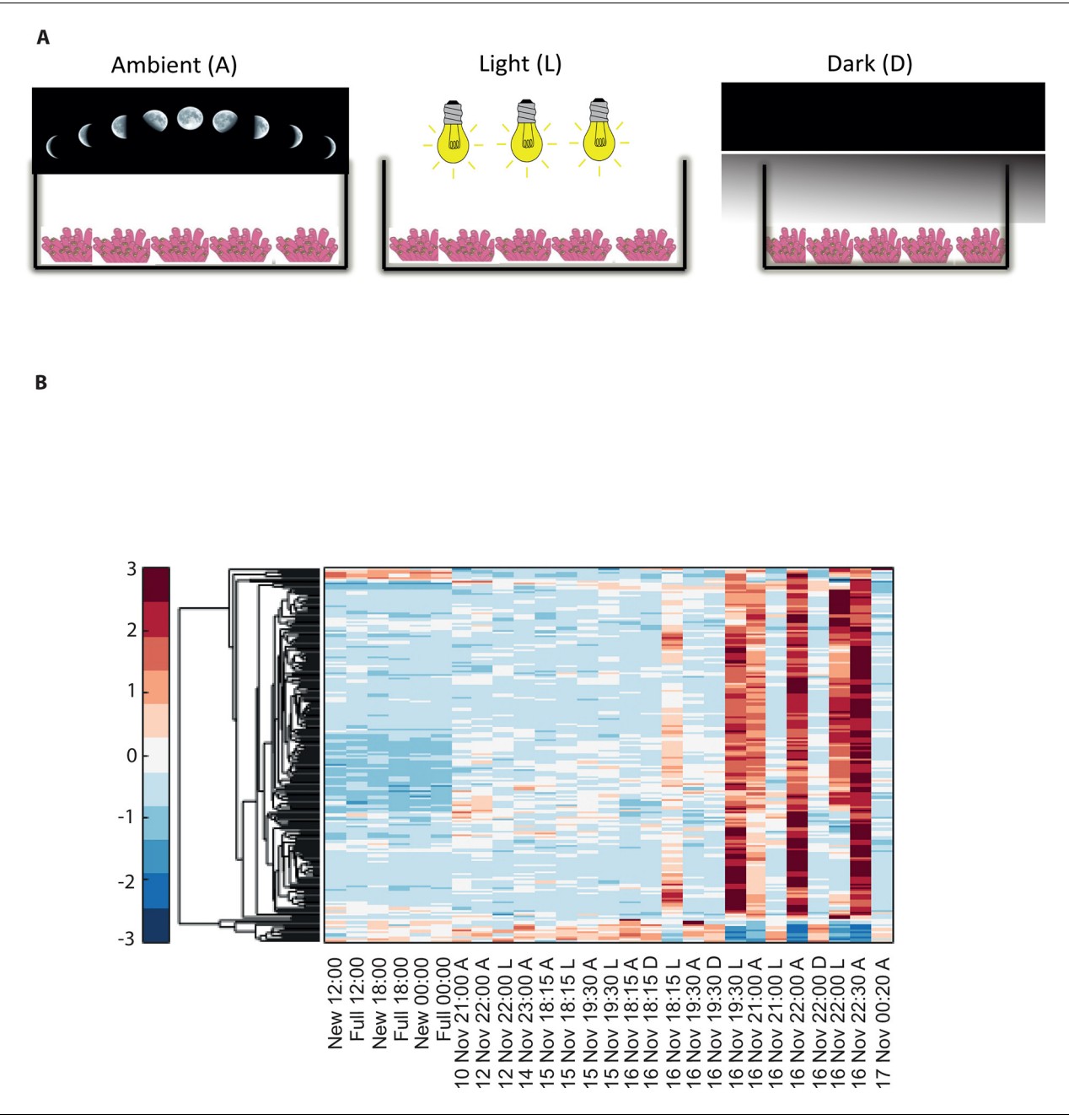

**Figure 2.** Disrupting synchronized mass spawning in the coral, *Acropora millepora*. (**A**) Beginning 8 days prior to the spawning night, *A. millepora* colonies were exposed to one of the following treatments: ambient (A), in which colonies were exposed to natural day and night cycles with full exposure to moon light; light (L), in which colonies were exposed to natural daylight during the day and artificial photosynthetically active radiation (PAR) light (~5 μmol quanta m$^{-2}$ s$^{-1}$) at sunset every day for ~6 hrs (between 18:15 and 24:00) and then left in the dark until sunrise; or dark (D), in which colonies were exposed to natural daylight during the day and left in the dark from 18:15 to sunrise. (**B**) Hierarchical clustering of *A. millepora* gene expression data for the 184 coral transcripts that were only variable during the spawning night, depicting gene expression changes between treatments A, L and D denote ambient, light and dark treatments, respectively.

The following figure supplements are available for Figure 2:

**Figure supplement 1.** Hierarchical clustering of *Acropora millepora* gene expression data for (**A**) the 177 coral transcripts that were up-regulated and (**B**) the 29 coral transcripts that were down-regulated during the spawning night.

*Figure 2. continued on next page*

*Figure 2. Continued*

**Figure supplement 2.** Correlation of gene expression Log2 fold change between values obtained from RNA-seq analysis and expression values obtained using quantitative PCR (qPCR).

**Figure supplement 3.** The effect of light quantity and quality on the timing of broadcast spawning in the coral *Acropora millepora* at Heron Island.

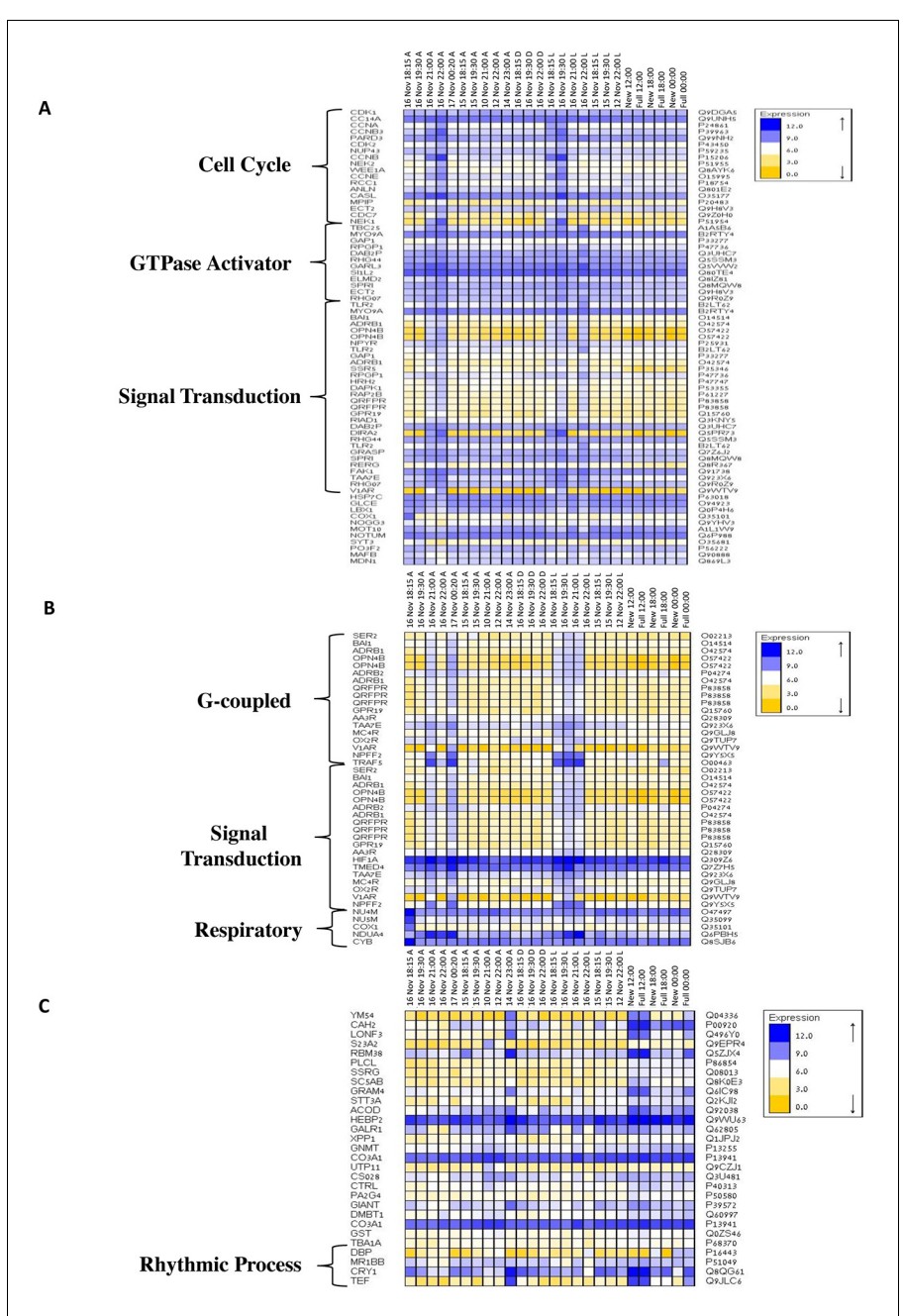

**Figure 3.** Gene ontology enrichment analysis for *Acropora millepora* gene variability during mass spawning. Gene enrichments (false discovery rate<0.1) across GO categories are shown. GOseq was used to test for enriched GO categories of genes that were (**A**) variable during spawning, (**B**) up-regulated during spawning or (**C**) down-regulated during spawning. The left and right identifiers represent the gene names and UniProt accession numbers, respectively.

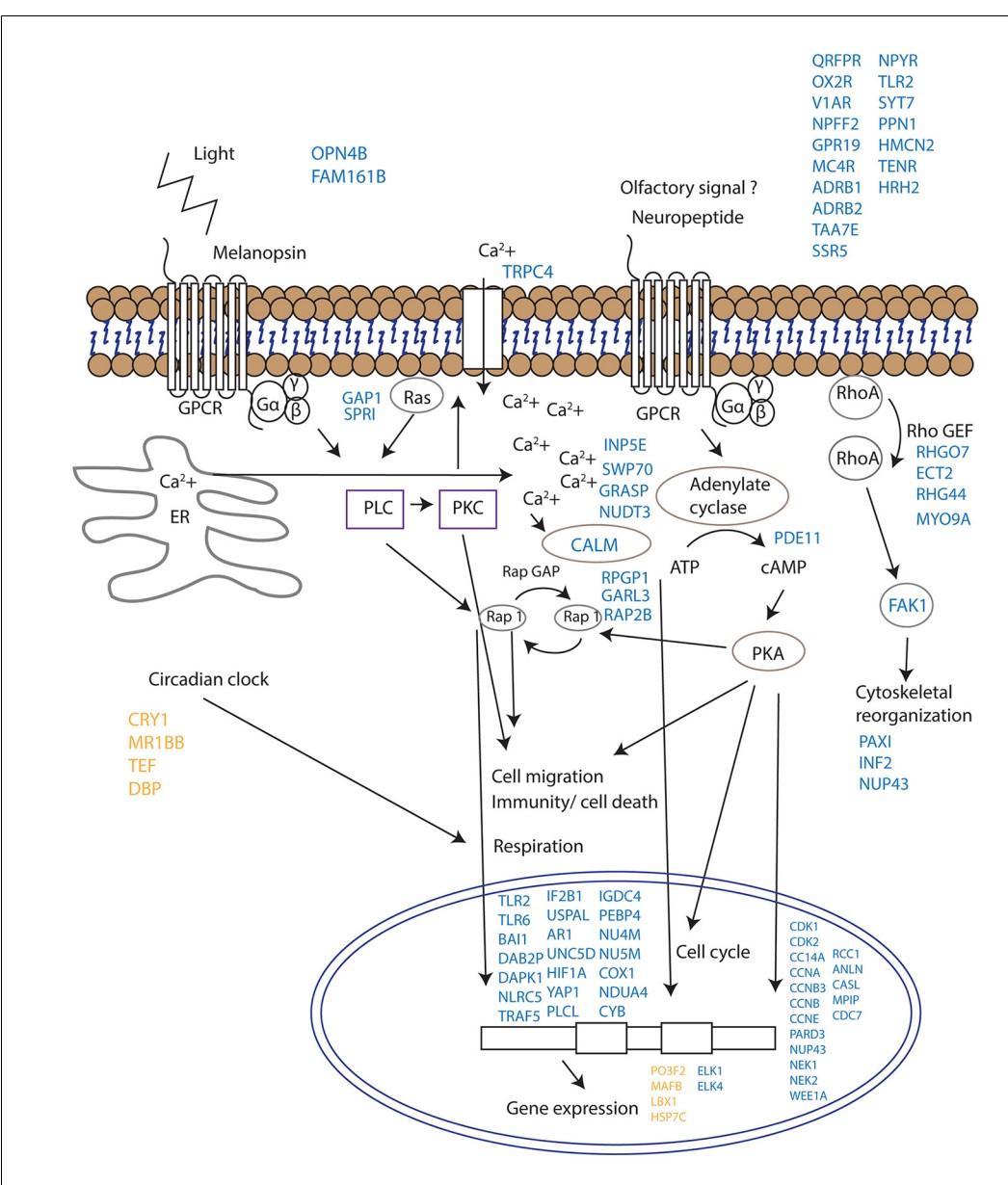

**Figure 4.** Proposed model for signalling events during spawning and gamete release in *Acropora millepora*. The release of gametes occurs in the presence of moonlight or another signal (such as an olfactory signal) that stimulates melanopsin-like homologs and/or other neuropeptides and commences GPCR signalling cascades upon receptor activation. This effect results in cell migration, follicle rupture, changes in immunity/cell death, respiration and cytoskeletal organization, and the combination of which results in a synchronized gamete release. Blue labels indicate genes being up-regulated; orange labels indicate genes down-regulated.

Our study does not provide direct evidence as to whether the melanopsin-like homologs that are up-regulated during spawning are true photopigments, as they have been shown to be in mammals (*Peirson et al., 2009*). The sequential expression pattern of these melanopsin-like homologs, however, suggests that they are important for signalling events during mass spawning in corals regardless of whether they convey a light, pheromone or other signal. The discovery of potential neuropeptide/GPCR-coupled signalling mechanisms are consistent with neurohormones playing a role in synchronized spawning events in tropical abalone (*York et al., 2012*) and in the settlement behavior of coral larvae (*Grasso et al., 2011*). Additionally, an increase in a tachykinin-like peptide receptor suggests that a pheromonal trigger may be involved (*Winther et al., 2006*). Given that the

transient receptor potential protein superfamily represents calcium permeable channels which have important roles in phototransduction in *Drosophila* and in mammalian vision (*Panda, 2002*; *Pan et al., 2011*), our results showing an up-regulation of Trp-related protein 4 fits with the proposed hypothesis that cytoplasmic calcium may act as a secondary messenger for coral photoreceptors (*Hilton et al., 2012*).

Although the cues for the synchronization of gamete release in broadcast spawning corals are complex, our results indicate that nocturnal illumination is an important factor and that changes in the light regime surrounding corals can desynchronize the timing of gamete release. In this respect, only colonies exposed to ambient light conditions at night spawned. We further tested the concept of light regime changes causing a mismatch in the timing of gamete release through another experiment in which *A. millepora* colonies were exposed to treatments where the length of the day was extended artificially (between sunset and midnight), through exposure to a suite of light quantity and quality regimes. We found that colonies exposed to ambient conditions and treatments with light sources in the red light region (620–700 nm) of the spectrum spawned at 21:30, which was synchronous with *A. millepora* colonies spawning in the field. However, colonies exposed to light in the green (500–620 nm) and blue (400–500 nm) regions of the spectrum and to PAR (400–700 nm) at low-dim, medium and high light intensities phase shifted their spawning time by a 6–8 hr delay or spawned 2 nights later as compared with colonies under ambient conditions (*Figure 2—figure supplement 3*). Our results suggest that detection of light in the blue and green parts of the spectrum is important for gamete release, which is congruent with suggestions that blue light detection is a key element for coral spawning behavior (*Gorbunov and Falkowski, 2002*) and evidence that *A. millepora* larvae settle more readily under blue and green light as compared with red light (*Strader et al., 2015*). Furthermore, our results from both experiments (*Figure 2*, *Figure 2—figure supplement 3*) on the effect of 'light pollution' on coral spawning behavior suggest that disruption or phase shift delay in spawning time can occur rapidly, that is, within 7 days of exposure to changes in nocturnal light regimes. These findings differ from previous studies which suggested that entrainment to moonlight rhythms occurs over several months (*Willis et al., 1985*).

Our results on the effects of light on the timing of spawning are crucial because sexual reproduction is one of the most important processes for the persistence of reefs. The interplay between endogenous clocks and external cues in an era of industrialization and global change (when artificial lights compete with moonlight to affect reproductive timing and fertilization success in broadcasting species) should be considered in plans to protect coral reefs and marine ecosystems.

## Materials and methods

### Coral collection and experimental design

Twenty whole colonies of *A. millepora* were collected on November 9, 2011 from the Heron Island reef flat (23 33'S, 151 54'E), GBR, Australia. Small branches were cut from the central portion of each colony prior to specimen collection to confirm pink-coloured eggs, which are present when a colony is reproductively mature (*Harrison et al., 1984*). Four of the colonies were transported close to the shore but were left in the field on the reef flat (treatment F). The remaining 16 colonies were transferred to large, outdoor flow-through aquaria and were exposed to natural sunlight, moonlight and flow-through seawater from the reef flat. Colonies in the outside aquaria were divided into the following three treatments: ambient (A), in which the colonies (N=6) were exposed to natural day and night cycles and full exposure to moonlight; light (L), in which the colonies (N=5) were exposed to natural daylight during the day and to artificial PAR light (~5 µmol quanta m$^{-2}$ s$^{-1}$) post sunset every day for ~6 hrs between 18:15 and 24:00 and then left in the dark until sunrise; and dark (D), in which the colonies (N=5) were exposed to natural daylight during the day and left in the dark between 18:15 and sunrise. The experiment was conducted at Heron Island Research Station in an area that was maintained in darkness at night to avoid artificial light contamination from non-experimental sources. Branches from colonies were collected from each of the three treatments on 4 d (November 10, 12, 14, and 15, 2011) leading up to the spawning night (November 16, 2011) (*Figure 2A*). The corals were sampled on these days at noon and during moonrise, which was between 21:00 and 23:00 depending on the day. On November 16 (the spawning night), we sampled the corals at noon, 18:15, 19:30, 21:00, 22:00, 22:30 and 00:20. The release of gametes

occurred between 21:30 and 22:30, and the colonies in the Ambient treatment began to show signs of setting at 19:30. We also sampled released gametes from colonies in the Ambient treatment. Sampled coral branches were snap-frozen in liquid nitrogen and stored at -80°C until processing for total RNA extraction. Additional branches were sampled (N=4) during August from the reef flat at a 2-m depth during new moon and full moon days at 12:00, 18:00, and 24:00.

## Total RNA isolation

Total RNA from the coral branches was isolated by homogenizing 100 mg coral tissue in 1 ml TRIzol (Invitrogen) according to the manufacturer's instructions. The RNA was then extracted once with 1 volume chloroform and precipitated in ½ volume isopropanol, washed in 1 volume of 75% ethanol and subsequently dissolved in RNase-free water. These samples were then processed through a 5 M LiCl precipitation overnight at −20°C, washed three times with 75% ethanol and subsequently dissolved in RNase-free water. The integrity and quality of the total RNA was assessed using a Bioanalyzer (Agilent Technology). Only the samples showing intact RNA (RNA integrity number >8) were used for the RNA-seq analysis.

## RNA-Seq analysis

The Illumina TruSeq protocol was used to prepare libraries from the RNA samples. Overall, 12 libraries of full moon and new moon samples were run on one lane and 24 libraries of samples from the spawning experiment were run on two additional lanes in the Illumina HiSeq2000 machine using the multiplexing strategy of the TruSeq protocol. On average, ~10 million single-end reads were obtained for each sample of the full moon and new moon conditions, and ~15 million paired-end reads were obtained for each sample in the spawning experiment. The sequencing data reported in this study was deposited to the Sequence Read Archive (SRA), under accession SRP055723. The reads were 100 bases long. TopHat (*Trapnell et al., 2009*) was used to align the reads against the *A. millepora* genome, keeping only uniquely aligned reads with up to two mismatches per read. Only reads that were aligned to the protein coding regions of the *A. millepora* genes (*Moya, 2012*) were used. A custom script written in Perl was used to parse the output of TopHat (given in the Sequence Alignment/Map (SAM) format [http://samtools.sourceforge.net/]) and to convert it into the raw number of reads aligned to each *A. millepora* gene (available via Dryad data repository: doi:10.5061/dryad.541g6). The *A. millepora* gene information was downloaded from the NCBI database, we retained only genes that were found to be significant similar (Blastx E-value<1e-6) to the Swiss-Prot proteins data set. On average, ~40% of the reads in the full moon and new moon conditions and ~30% of the reads of the spawning conditions passed all filters and were mapped to the coding regions of the 12,384 *A. millepora* genes with Swiss-Prot homology. These numbers are comparable to the ones obtained by a similar analysis in a better annotated animal, the zebrafish, in which between 10% to 56% of the reads mapped to the coding regions of known genes (*Ben-Moshe at al., 2014*; *Tovin et al., 2012*).

## Comparison of transcriptome under full versus new moon

Statistically significant differences between the number of reads aligned to each *A. millepora* gene (the expression profile) in the tested conditions were identified as described (*Alon et al., 2011*). Briefly, the expression profiles were normalized using a variation of the TMM normalization method (*Robinson and Oshlack, 2010*). Subsequently, we searched for expression differences between samples associated with the full versus new moon conditions that could not be explained by the expected Poisson noise with a p-value <0.05 and using a Bonferroni correction for multiple testing (*Alon et al., 2011*). Each one of the full moon and new moon samples was sequenced twice (technical duplicates), and as the correlation between the gene's expression levels in these duplicates was very high (average Pearson's correlation >0.999) the average expression level is presented in the figures.

## Clustering analysis

The data from the full moon and new moon experiments and the spawning experiment were combined and normalized as described above. We then performed a hierarchical clustering (MATLAB, MathWorks) of all of the genes that (1) had expression levels above the median of all of the gene

expression levels in at least one sample and (2) had a variance in the expression level that was above the median variance of all of the genes. The gene's variance was normalized by the gene's mean expression level (the expected variance assuming Poisson noise). Overall, 4756 genes out of the original list of 12,384 genes passed these two filters and were therefore considered to be both expressed and to have an expression profile that was not constant. Clustering of the expression patterns of these genes indicates that a sizable number of genes are highly variable only on the spawning day (*Figure 1—figure supplement 2*). These genes are characterized and identified below.

## Spawning analysis

The genes that were highly variable during the spawning day in ambient light conditions (regular light-dark cycles), which ultimately led to the spawning event, could be responding to at least two factors: (1) moonlight (either directly via light or indirectly through the time of the month) or (2) the time of day. Alternatively, these genes can also be variable on other days, such as the days prior to or after the spawning day. Finally, these highly variable genes could also be part of the molecular mechanism of spawning. To identify the latter group, we selected all of the genes that were highly variable during the spawning day in ambient conditions but significantly less variable in the full moon and new moon conditions at different times of the day (without spawning events) and the days prior to the spawning events (all under ambient conditions).

Variability was estimated using a gene's variance divided by its expected variance assuming Poisson noise in the gene expression levels, which is equal to the gene's mean expression level. All of the gene variability measures were sorted, and we selected genes that were in the top 10% on the spawning day (i.e., highly variable) but not in the top quartile (i.e., not as variable) in the previous days and in the full moon and new moon experiments. We noted that changing the cut-offs produced similar results, for example by selecting genes that were in the top 5–25% on the spawning day but not in the top 25–50% in the previous days and in the full moon and new moon experiments. Additionally, we pursued high top expression levels (more than 100 reads) of these genes in any one of the mentioned conditions (only one quarter of the genes showed this expression level or higher).

We also identified genes with higher or lower expressions on the spawning day compared with the previous days and those of the full moon and new moon experiments. We selected for at least a twofold difference between the mean expression level on the spawning day and (A) the mean expression on the previous days, as well as (B) the mean expression in the full moon and new moon experiments. Thus, the difference in expression level had to be at least twofold in respect to both (A) and (B). Additionally, we filtered for greater than 100 reads in the top expression level in the higher condition.

The genes detected were analyzed using the software GOseq (*Young et al., 2010*) to find statistically significant over-represented functional groups with a Benjamini-Hochberg false discovery rate equal to or less than 0.1. We compared the gene's expression pattern in the described conditions with the expression pattern of the released gametes that were sequenced, and found no significant correlation between the gene's expression patterns (average Pearson's correlation of 0.22 between the released gametes and all the other described conditions). Specifically, for the genes detected above as having higher or lower expression on the spawning day, the correlation with the released gametes sample was even lower (average Pearson's correlation of 0.05 between the released gametes and all the spawning day conditions).

## QPCR validation

The expression patterns of the RNA-seq data for selected genes with variable expressions during spawning (*Supplementary file 1*) were validated via quantitative PCR (qPCR). Primers were designed from RNA-seq data using the Primer Express Software v3.0 (Applied Biosystems, USA). Total RNA (1000 ng) was reverse transcribed with a SuperScript VILO cDNA Synthesis Kit (Invitrogen) following the manufacturer's instructions. Transcript levels were determined via qPCR assays using an Eppendorf 5075 (Applied Biosystems, USA) robot to dispense SYBR Green PCR master mix (Applied Biosystems, UK) into 384-well plates, and assays were run in a 7900HT Fast Real-time PCR System (Applied Biosystems, USA). The PCR conditions included an initial denaturation for 10 min at 95°C, followed by 45 cycles of 95°C for 15 s and 60°C for 1 min. Finally, a dissociation step included 95°C for 2 min, 60°C for 15 s and 95°C for 15 s. The final reaction volume was 10 μl and included

300 nM of primers. All reactions were carried out with two technical replicates (which showed no difference in expression levels). For each candidate gene sample (from five replicates), the ambient colonies at 19:30 were tested against the ambient colonies at 22:00 on the spawning night. A no-template control and a no-reverse transcription control were performed for each gene and treatment to ensure that the cDNA samples and PCR reagents did not undergo DNA contamination. Additionally, to ensure the specificity of the primers for the coral cDNA, they were tested on cDNA and genomic DNA from *Symbiodinium* sp. as a template in a PCR. This procedure avoided any amplification of non-coral DNA. The comparative delta CT method (*Walker, 2002*) and a maximal PCR efficiency for each gene (E=2) were used to determine the relative quantities of mRNA transcripts from each sample. Each value was normalized to two reference genes, adenosylhomocysteinase (AdoHcyase) and ribosomal protein L7 (Rpl7). The selection of reference genes for this experiment was performed using a pool of reference genes (*Supplementary file 3*) and analysing their expression stability via the geNorm software (*Vandesompele et al., 2002*). For this study, the most stable expression was found for AdoHcyase and Rpl7 (M value=0.225), and a minimum of two reference genes was recommended (V2/3=0.102). The real-time dissociation curve was used to check for the presence of a unique PCR product. Relative expression values for each gene were calculated by showing a ratio of the relative expression at 22:00 over the relative expression at 19:30. The results of the qPCR and RNA-seq analyses are presented on a $\log_2$ scale. A Welch t-test was used to test for significant differences in the expression levels (obtained via quantitative real-time PCR) of each gene at 19:30 compared with that at 22:00 (*Supplementary file 3*).

## Effect of light quantity and quality on mass spawning synchronization

Ninety healthy colonies of *A. millepora* (which were checked for maturity as described above) were transferred from the Heron Island reef flat and placed in the outdoor flow-through aquaria which was continuously flushed with seawater obtained from the Heron Island reef flat. Five colonies were placed into each aquarium, which were shaded during the day to simulate PAR levels at a 3-m depth. The coral colonies were acclimated to ambient seawater temperatures for 48 hrs prior to the experiment. At night, the shading was removed after sunset to expose the corals to natural moonlight. We simulated later 'sunsets' to determine if light intensity and light spectra can phase shift-spawning time. The coral colonies were divided between aquaria exposed to the light intensity treatments (n=15) and light spectra treatments (n=10); the coral colonies were divided into three aquaria for each of the seven treatments. The coral colonies were irradiated with artificial light every day for 6 hrs between 18:00 and 24:00 (midnight). These colonies also received the ambient light/dark cycle during the day. We tested three light intensities (100 ['high'], 50 ['medium'] and 0.75–1 ['low-dim'] µmole quanta $m^{-2}$ $s^{-1}$ [using T8 fluorescent lamps]) and three light spectra (blue [400–500 nm, Lee filter model, deep blue 120 40% transmission], green [500–620 nm, Lee filter model, dark green 124 60% transmission] and red [620–700 nm, Lee filter model, bright red 026, 80% transmission]). The light intensity was adjusted to 1 µmol quanta $m^{-2}$ $s^{-1}$ for all three light spectra using a neutral density filter (Lee filter model, neutral density 210 (0.6) 20% transmission) when needed. The light was measured using a LI-COR (LI-192S) light meter. As a control, we maintained another set of corals that were exposed to ambient light/dark cycles. The corals were divided into four large tables; the first table included the control tanks, the second the 'high' light intensity treatment, the third the 'medium' light and the fourth table the 'low' light treatment corals. At the ends of each table, we added another four tanks into which were projected the blue, green and red light spectra. To avoid any light 'leaks', black plastic screens were placed between the treatments. The experiment was conducted on a roof deck at the Heron Island Research Station to avoid artificial light 'contamination' and increase exposure to moonlight when the corals were not artificially irradiated. The experiment began on October 31, 2006 and ended on November 15, 2006. The corals were monitored at night each day after the full moon (November 5, 2006); moonrise began at 17:53, and moonset was at 04:22 (November 6). Each night, the corals were monitored between 18:00 and 03:00 or until spawning ceased. The times when the gamete bundles appeared in the polyp mouths ('setting') were recorded, as well as the times at which the first and last gamete bundles were released into the water column. The data were analyzed based on the timing of gamete release. Spawning in the field for the same species was observed via snorkeling or scuba diving. The major spawning on the reef and in our control aquaria occurred on the night of November 13, 2006, which was 8 nights after the full moon night. After the spawning event on November 13, the corals were held in the aquaria and

monitored for later spawning in the ambient light/dark cycles with no artificial light until November 15. At the end of the experiment, the coral colonies were returned to the Heron Island reef flat.

## Acknowledgements

The authors thank A Tarrant for comments on drafts of the manuscript and Heron Island Research Station staff for their help during experiments. This work was supported by the Australian Research Council (PK and OHG). The sequencing data reported in this study has been deposited to the Sequence Read Archive (SRA), under accession SRP055723. The authors also thank Prof. P Harrison for his supportive review and the two additional anonymous reviewers in improving this manuscript.

## Additional information

### Funding

| Funder | Author |
| --- | --- |
| Australian Research Council | O Hoegh-Guldberg |

The funders had no role in study design, data collection and interpretation, or the decision to submit the work for publication.

### Author contributions

PK, Conception and designs, Acquisition of data, Analysis and interpretation of data, Drafting or revising the article, Contributed unpublished essential data or reagents; SA, Acquisition of data, Analysis and interpretation of data, Drafting or revising the article; SKL, Analysis and interpretation of data; OHG, Drafting or revising the article, Contributed unpublished essential data or reagents; OL, Conception and design, Acquistion of data, Analysis and interpretation of data, Drafting or revising the article, Contributed unpublished essential data or reagents

## Additional files

### Supplementary files

• Supplementary file 1. *Acropora millepora* genes up-regulated during broadcast spawning, genes also detected to be variable during the spawning day are highlighted. The time of tides and moon-rise during November 2011 spawning event.

• Supplementary file 2. Gene ontology enrichment analysis for *Acropora millepora* gene variability between new moon and full moon days. Gene enrichments (false discovery rate<0.1) across GO categories are shown. GOseq was used to test for enriched GO categories.

• Supplementary file 3. List of candidate genes used in qPCR expression analysis.

### Major datasets

The following datasets were generated:

| Author(s) | Year | Dataset title | Dataset ID and/or URL | Database, license, and accessibility information |
| --- | --- | --- | --- | --- |
| Kaniewska P, Alon S, Karako-Lampert S, Hoegh-Guldberg O, Levy O | 2015 | Data from: Unwinding the mystery of coral broadcast mass spawning, signaling cascades and the importance of moonlight | http://dx.doi.org/10.5061/dryad.541g6 | Available at Dryad Digital Repository under a CC0 Public Domain Dedication |
| Kaniewska P, Alon S, Karako-Lampert S, Hoegh-Guldberg O, Levy O | 2015 | Acropora Raw RNA Sequencing Data | http://www.ncbi.nlm.nih.gov/sra/?term=SRP055723 | Publicly available at the NCBI Short Read Archive (Accession no: SRP055723). |

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
