## [Decision Letter]

Thank you for submitting your work entitled "Unwinding the mystery of coral broadcast mass spawning, signaling cascades and the importance of moonlight" for peer review at *eLife*. Your submission has been favorably evaluated by Ian Baldwin (Senior Editor) and three reviewers, one of whom served as Guest Reviewing Editor. One of the three reviewers, Peter Harrison, has agreed to reveal his identity.

The reviewers have discussed the reviews with one another and the Reviewing Editor has drafted this decision to help you prepare a revised submission.

Summary:

In their manuscript, Kaniewska et al. use experimental manipulations and transcriptomics to examine the signaling mechanisms underlying coordinated spawning of *A. millepora*. The authors provide evidence indicating that excessive light delays or prevents spawning. They also identify transcriptional changes during spawning that suggest molecular pathways that likely regulate spawning events. In conclusion, this is an important article that lays the foundation for future studies, including comparative studies with closely related species (2 *acroporas*, for example) whose time of spawning is different, and thus may allow finding the precise mechanism of time to spawning. The work is carefully performed and worthy of publication once the following revisions, indicated by the reviewers and outlined below, have been addressed.

Essential revisions:

1) The authors suggest that red light is the most important spectrum involved for sperm release. However, Gurbunov and Falkowski (2002) have suggested that detection of blue spectrum of moonlight could be the key to spawning. The authors should comment on this apparent discrepancy in the Discussion.

2) Figure 2—figure supplement 3 should be included in the manuscript. It is discussed in the text, represents a substantial experimental undertaking, and has important implications. This figure could easily be adapted to one panel by simply placing the blue line (phase shift) of panel B into panel A. A legend defining the shaded regions would add clarification. To include this additional figure, I think Figure 1 and Figure 2 could be consolidated into one figure, with one panel describing the changes in transcript expression during "undisrupted" spawning and the lower panel describing the changes in transcript expression following experimental disruption. I'm not sure that Figure 1 panel A (current iteration) is necessary.

3) Another interesting aspect of this study, which is not highlighted in the text, is that the results indicate that the disruption to spawning cycles is relatively rapid, such that within 7 days major changes can be induced. In contrast, some of the earlier studies suggested that moonlight rhythms might be entrained over at least a few months e.g. Willis et al. 1985. The authors should comment on this apparent discrepancy in the Discussion.

---

## [Author Response]

*Essential revisions:*

1) The authors suggest that red light is the most important spectrum involved for sperm release. However, Gurbunov and Falkowski (2002) have suggested that detection of blue spectrum of moonlight could be the key to spawning. The authors should comment on this apparent discrepancy in the Discussion.

There seems to be a misunderstanding, as we do not suggest in our text that red light is the most important spectrum involved for gamete release. In fact, our results suggest the opposite, that blue light is important, which fits with what was suggested by Goburnov and Falkowski 2002, and a recent study which investigated *A. millepora* coral larvae settlement showing preference for blue and green light of the. We have clarified this in the text now and have also added the suggested citation.

This text was revised accordingly, and we added the following passage: “Our results suggest that detection of light in the blue and green parts of the spectrum is important for gamete release, which is congruent with suggestions that blue light detection is a key element for coral spawning behaviour (Gorbunov and Falkowski, 2002) and evidence that *A. millepora* larvae settle more readily under blue and green light as compared to red light (Strader, Davies and Matz, 2015). Furthermore, our results from both experiments (Figure 2, Figure 2—figure supplement 3) on the effect of “light pollution” on coral spawning behaviour suggest that disruption or phase shift delay in spawning time can occur rapidly, that is, within 7 days of exposure to changes in nocturnal light regimes”.

*2) Figure 2—figure supplement 3 should be included in the manuscript. It is discussed in the text, represents a substantial experimental undertaking, and has important implications. This figure could easily be adapted to one panel by simply placing the blue line (phase shift) of panel B into panel A. A legend defining the shaded regions would add clarification. To include this additional figure, I think Figure 1 and Figure 2 could be consolidated into one figure, with one panel describing the changes in transcript expression during "undisrupted" spawning and the lower panel describing the changes in transcript expression following experimental disruption. I'm not sure that Figure 1 panel A (current iteration) is necessary.*

We argue that Figure 2—figure supplement 3 is better left as a supplemental figure as it describes a separate experiment that investigated the effect of light pollution on coral spawning behavior. If listed with the main figures, it might add confusion to the readers of *eLife*. The initiative of Figure 2—figure supplement 3 is only to support the partial finding of light pollution affecting the spawning. The supportive figure is describing a physiological experiment that was conducted during 2006-2007, and we are afraid it will be shadowing our main work from the 2011 experiment that includes physiological and RNseq sampling. However, if the editor insists on this change we will change this and include the figure as a main figure.

*3) Another interesting aspect of this study, which is not highlighted in the text, is that the results indicate that the disruption to spawning cycles is relatively rapid, such that within 7 days major changes can be induced. In contrast, some of the earlier studies suggested that moonlight rhythms might be entrained over at least a few months e.g. Willis et al. 1985. The authors should comment on this apparent discrepancy in the Discussion.*

As suggested, we have added a sentence to the Discussion describing the discrepancy between our results and that of Willis et al. 1985 (“These findings differ from previous studies which suggested that entrainment to moonlight rhythms occurs over several months)”.